# Phylogeny and Diversity of the Genus *Pseudohydnum* (Auriculariales, Basidiomycota)

**DOI:** 10.3390/jof8070658

**Published:** 2022-06-23

**Authors:** Hong-Min Zhou, Hong-Gao Liu, Genevieve M. Gates, Fang Wu, Yu-Cheng Dai, Jerry A. Cooper

**Affiliations:** 1Institute of Microbiology, School of Ecology and Nature Conservation, Beijing Forestry University, Beijing 100083, China; zhouhongmin11@bjfu.edu.cn (H.-M.Z.); fangwubjfu2014@bjfu.edu.cn (F.W.); 2School of Agronomy and Life Sciences, Zhaotong University, Zhaotong 657000, China; honggaoliu@126.com; 3Tasmanian Institute of Agriculture, Private Bag 98, Hobart, TAS 7001, Australia; genevieve.gates@utas.edu.au; 4Manaaki Whenua, Landcare Research, P.O. Box 69040, Lincoln 7608, New Zealand

**Keywords:** gelatinous fungi, taxonomy, wood-inhabiting fungi

## Abstract

The toothed jelly fungus *Pseudohydnum gelatinosum* was originally described from Europe. The name has a broad sense and the species has been widely reported almost all over the world. We have studied samples of so-called *P. gelatinosum* from Asia and Oceania. Based on morphology, hosts, geography, and phylogenetic analysis using the internal transcribed spacer regions (ITSs) and the large subunit of nuclear ribosomal RNA gene (nLSU), four new species, *P. himalayanum, P. orbiculare, P. sinogelatinosum,* and *P. tasmanicum,* from China, New Zealand, and Australia are described and illustrated, and a new combination, *Pseudohydnum totarae*, is proposed. The five new taxa can be differentiated by the shape of their basidiomata, pileal surface color when fresh, spine size, basidiospore dimensions, shape of hyphidia, hosts, and biogeography. Phylogenetically, most of these taxa are distantly related, and different base pairs among these taxa mostly account for >2% nucleotides in the ITS regions.

## 1. Introduction

*Pseudohydnum* P. Karst. is an easily recognizable genus in the Auriculariales, characterized macroscopically by flabellate or conchate and gelatinous basidiomata with a hydnoid hymenophore, and microscopically by longitudinally cruciate-septate basidia [1,2]. The genus was classified in the Tremellales because of its longitudinally septate basidia [2,3]. Bandoni considered *Pseudohydnum* a member of the Auriculariales based on micromorphological, ultrastructural, ecological, and developmental data, and this has been confirmed with phylogenetic analysis [4,5,6].

Currently, three species of *Pseudohydnum*, *P*. *brunneiceps* Y.L. Chen et al., *P*. *gelatinosum* (Scop.) P. Karst., and *P*. *translucens* Lloyd, are accepted [1,7,8]. In addition, two forms and two varieties of *P*. *gelatinosum* are recognized: *P*. *gelatinosum* f. *album* (Bres.) Kobayasi and *P*. *gelatinosum* f. *fuscum* (Bres.) Kobayasi from Europe and Japan [9,10], *P*. *gelatinosum* var. *bisporum* Lowy & Courtec. from French Guiana [11], and *P*. *gelatinosum* var. *paucidentatum* Lowy from Bolivia [12,13].

*Pseudohydnum guepinioides* Rick and *P*. *thelephorum* (Lév.) Lloyd were described from Brazil [14] and French Guiana [15], respectively, but both of them are considered synonyms of *Trechispora thelephora* (Lév.) Ryvarden [16].

The ‘toothed jelly’ fungus *Pseudohydnum gelatinosum* is the type species of the genus and was originally described from Croatia in Europe as *Hydnum gelatinosum* Scop. [17]. The name is applied broadly and the species has been widely reported from Europe [3,18,19], North America [20,21,22], Central and South America [2,11,13,23], Asia [24,25,26], and Oceania [27,28,29,30,31]. The fungus is considered to be edible as well as medicinal, with antitumor and antioxidant properties [21,22,32], and a recent study has shown the extract from *P*. *gelatinosum* to have antimicrobial activity [33].

Recently, more samples of so-called *Pseudohydnum gelatinosum* were collected from Asia and Oceania, and the aim of the present study is to clarify the species diversity of *Pseudohydnum* in the two land masses and outline a phylogeny of *Pseudohydnum* based on the known data from over the world.

## 2. Materials and Methods

### 2.1. Morphological Studies

The studied specimens are deposited in the herbaria of the Institute of Microbiology, Beijing Forestry University (BJFC) and Landcare Research (PDD), New Zealand. Morphological descriptions are based on field notes and voucher specimens. Color terms are from Anonymous [34], Petersen [35], and Kornerup and Wanscher [36]. Microscopic structures refer to Pippola and Kotiranta [37], Malysheva et al. [38], and Fan et al. [39]. Handmade sections of voucher basidiomata were examined using a Nikon Eclipse 80i microscope (magnification ×1000) after being mounted in 5% KOH for five minutes and treated with 1% Phloxine B (C_20_H_4_Br_4_Cl_2_K_2_O_5_). Microscopic structures were photographed using a Nikon Digital Sight DS-L3 or Leica ICC50 HD camera. At least 20 basidia and basidiospores from each specimen were measured. Spore measurements are quoted as means and standard deviations (±1.5 σ) or with 5% of measurements excluded from each end of the range given in parentheses. Stalks were excluded for basidia measurements, and the hilar appendages were excluded for basidiospore measurements. The following abbreviations are used in the descriptions: IKI = Melzer’s reagent, IKI– = neither amyloid nor dextrinoid, CB = Cotton Blue, CB– = acyanophilous in Cotton Blue, L = arithmetic average of all spore lengths, W = arithmetic average of all spore widths, Q = L/W ratios, *n* (a/b) = number of spores (a) measured from a given number (b) of specimens, and σ = standard deviation from the mean.

### 2.2. DNA Extraction, Amplification, and Sequencing

A CTAB rapid plant genome extraction kit-DN14 (Aidlab Biotechnologies Co., Ltd., Beijing, China) or DNA easy Plant Mini Kit (Aidlab Biotechnologies, Beijing, China) was used to obtain DNA from dried specimens and perform a polymerase chain reaction (PCR) according to the manufacturer’s instructions, with some modifications [40,41]. Two DNA gene fragments, i.e., the internal transcribed spacer (ITS) and large subunit nuclear ribosomal RNA gene (nLSU), were amplified using the primer pairs ITS5/ITS4 [42] and LR0R/LR7 (https://sites.duke.edu/vilgalyslab/rdna_primers_for_fungi/, accessed on 22 April 2022) [43]. The PCR procedure for ITS was as follows: initial denaturation at 95 °C for 3 min, followed by 35 cycles at 94 °C for 40 s, 54 °C for 45 s, and 72 °C for 1 min and a final extension of 72 °C for 10 min. The PCR procedure for nLSU was as follows: initial denaturation at 94 °C for 1 min, followed by 35 cycles at 94 °C for 30 s, 50 °C for 1 min, and 72 °C for 1.5 min and a final extension of 72 °C for 10 min. DNA sequencing was performed at the Beijing Genomics Institute (BGI) and Manaaki Whenua, Landcare Research, Auckland. All newly generated sequences have been submitted to GenBank and are listed in Table 1.

### 2.3. Molecular Phylogenetics Analysis

The concatenated ITS + nLSU dataset included 34 ITS sequences and 22 nLSU sequences from 34 samples representing twelve taxa. The dataset had an aligned length of 1942 characters, of which 1521 were constant, 107 were variable and parsimony-uninformative, and 294 were parsimony-informative. MP analysis yield a tree (TL = 690, CI = 0.799, RI = 0.915, RC = 0.721, HI = 0.201). The best model for the concatenated ITS + nLSU dataset estimated and applied in the BI analysis was ‘GTR + F + I + G4’, lset nst = 6, rates = invgamma; prset statefreqpr = dirichlet (1,1,1,1).

The sequences generated for this study were aligned with additional sequences downloaded from GenBank. Both ITS and nLSU sequences were aligned using MAFFT v. 7 online (https://mafft.cbrc.jp/alignment/server/, accessed on 22 April 2022), adjusting the direction of nucleotide sequences according to the first sequence (accurate enough for most cases) and selecting the G-INS-i iterative refinement method [44]. Alignments were manually adjusted to maximum alignment and minimize gaps by BioEdit v. 7.0.9 [45]. A dataset composed of concatenated ITS + nLSU sequences was used to determine the phylogenetic position of the new species. *Protomerulius subreflexus* (Lloyd) O. Miettinen & Ryvarden and *Protomerulius substuppeus* (Berk. & Cooke) Ryvarden were selected as the outgroups, following Chen et al. [8].

Maximum Parsimony (MP) analysis was used for the dataset in PAUP* v. 4.0b10 [46]. All characters were equally weighted, and gaps were treated as missing data. A heuristic search option was used to infer the trees with TBR branch swapping and 1000 random sequence additions. All parsimonious trees were saved. The max-trees were set to 5000. Branches were collapsed if the minimum branch length was zero. Clade robustness was assessed using bootstrap analysis with 1000 replicates [47]. Descriptive Tree Statistics Tree Length (TL), Consistency Index (CI), Retention Index (RI), Rescaled Consistency Index (RC), and Homoplasy Index (HI) were calculated for each maximum parsimonious tree.

The CIPRES Science Gateway was used for Maximum Likelihood (ML) analysis based on the dataset in [48]. All parameters in the ML analysis used default settings, and statistical support values were obtained using nonparametric bootstrapping with 1000 replicates.

Bayesian Inference (BI) analysis based on the dataset was carried out using MrBayes v3.2.6 [49]. The best substitution model for the datasets was selected by ModelFinder [50], and trees were sampled every 100 generations. Four Markov chains were run from random starting trees for 2 million generations. Trees were sampled every 1000th generation. The first 25% of sampled trees were discarded as burn-in, whereas other trees were used to construct a 50% majority consensus tree and for calculating Bayesian posterior probabilities (BPPs).

## 3. Results

### 3.1. Molecular Phylogeny

BI analysis yielded a similar topology to MP and ML analysis, with an average standard deviation of split frequencies = 0.004227. Only the MP tree is provided here (Figure 1). Branches that received bootstrap support for MP (MP-BS), ML (ML-BS), and BI (BPP) greater than or equal to 70% (MP-BS and ML-BS) and 0.90 (BPP) were considered as significantly supported, respectively.

The current phylogeny placed all samples of *Pseudohydnum* in a fully supported clade (Figure 1). Two previously accepted species, *P*. *brunneiceps* and *P*. *gelatinosum*, received strong support. The samples from North America are treated as “*Pseudohydnum gelatinosum*-1”, “*Pseudohydnum gelatinosum*-2”, and “*Pseudohydnum gelatinosum*-3”. Nine Chinese specimens formed two distinct lineages, and seven Oceanian samples formed three distinct linages nested in *Pseudohydnum*.

### 3.2. Taxonomy

***Pseudohydnum himalayanum*** Y.C. Dai, F. Wu & H.M. Zhou, sp. nov. Figure 2.

Mycobank number: 844487.

**Type**—China. Yunnan Province, Lijiang, Yulong County, Yulong Snow Mountain, on rotten wood of *Abies*, 27.253333°, 100.174444°, elev. 3200 m, 16 September 2018, Cui 17045 (holotype, BJFC030344).

**Etymology**—***Himalayanum*** (Lat.): refers to the species being found in the east Himalayan area.

**Diagnosis**—Differs from other *Pseudohydnum* species by stipitate and confluent basidiomata, clay-pink to cinnamon pileal surface when fresh, dense spines 5–6 per mm at base, growth on *Abies* in high altitude (>3000 m), and occurrence in Southwest China.

**Basidiomata**—Annual, gelatinous when fresh, brittle when dry, with a lateral stipe, occasionally confluent sharing a stipe base. Pilei flabelliform, projecting up to 2.5 cm, 3.8 cm wide and 0.3 mm thick when dry. Pileal surface clay-pink (6D4) to cinnamon (6D6) when fresh, becoming purplish/date-brown when dry. Spines white and conical when fresh, become grayish brown when dry, 5–6 per mm at base, up to 0.2 mm long. Context translucent when fresh. Stipe concolorous with pileal surface, translucent when fresh, up to 2.5 cm long and 3 mm in diam. when dry.

**Hyphal structure**—Monomitic; generative hyphae with clamp connections. Contextual hyphae hyaline, thin- to slightly thick-walled, frequently branched, interwoven, 5–12 μm in diam. Tramal hyphae hyaline, thin-walled, frequently branched, interwoven, 1.5–2.5 μm in diam. Hyphidia simple. Basidia four-celled, barrel-shaped, ovoid to subglobose or globose, bearing guttules, 12–17.5 × 6–13.5 μm; sterigmata up to 12 μm long and 2–3.5 μm in diam. Probasidia fusiform to lageniform, 18–23 × 8–9 μm. Basidiospores broadly ellipsoid to subglobose, hyaline, thin-walled, occasionally with a guttule, IKI–, CB–, (6.5–)7–8.5 (–9) × (5.5–)6–7.2 (–8) μm, L = 7.75 μm, W = 6.55 μm, Q = 1.14–1.22 (*n* = 90/3).

**Habitat**—The species is rather common on rotten wood of *Abies* at altitudes higher than 3000 m.

**Distribution**—Distributed in mountain areas with coniferous forests of Southwest China.

**Additional specimens examined (paratypes)**—China. Tibet, Bomi County, the first tunnel from Bomi to Motuo, on fallen trunk of *Abies*, 29.783889°, 95.697222°, elev. 3500 m, Dai 23554 (BJFC038126); Yunnan Province, Lijiang, Yulong County, Ninety-nine Longtan, on rotten wood of *Abies*, elev. 3200 m, 15 September 2018, Cui 17030 (BJFC030329); Yulong Snow Mountain, on rotten wood of *Abies*, 27.253333°, 100.174444°, elev. 3200 m, 15 September 2018, Cui 17035 (BJFC030334); on fallen trunk of *Abies*, 16 September 2018, Cui 17065 (BJFC030364).

***Pseudohydnum orbiculare*** J.A. Cooper, sp. nov. Figure 3.

Mycobank number: 844488.

**Type**—New Zealand. South Island, West Coast Region, Ship Creek, Haast, on an unidentified rotten branch, −43.759431°, 169.149383°, 5 May 2018, NS2608 (holotype, PDD 112653).

**Etymology**—***Orbiculare*** (Lat.): refers to the species having orbicular basidiomata.

**Diagnosis**—Differs from other *Pseudohydnum* species by pileate basidiomata without a well-formed pseudostipe, orbicular pilei, dark-gray to black pileal surface when fresh, hymenophore with sparse spines (0.5–1 per mm at base), growth mostly on angiosperm, and distribution so far restricted to New Zealand.

**Basidiomata**—Annual, gelatinous when fresh, pileate, not confluent, in grouped tiers. Pilei laterally attached, with a narrow point of attachment, orbicular to dimidiate, up to 50 mm diam., pileal surface white to grayish brown (7E3–9E3) to reddish brown (11E5), smooth to minutely velutinate, with scattered, bluntly conical spines up to 0.2 mm long. Hymenophore white to pale-brown (8B2). Hymenophore spines white, conical, gelatinous, 2–4 mm long, and 1–2 mm wide at the base, 0.5–1 per mm at base. Context 0.5–1.5 mm thick, concolorous with pileal surface.

**Hyphal structure**—Monomitic; generative hyphae with clamp connections. Contextual and spine tramal hyphae hyaline, thin- to slightly thick-walled, frequently branched, agglutinate,1.5–9 µm in diam. Basidia longitudinally cruciate-septate, 10–14 × 10 µm, with four sterigmata up to 10 × 2 µm. Probasidia lageniform to clavate with stalks 1.5–2 µm in diam. Hyphidia irregular, mostly unbranched, slightly swollen towards apex, clamped at the base. Basidiospores (excluding apiculus) broadly ellipsoid to subglobose, hyaline, thin-walled, occasionally with a guttule, germinating by repetition, IKI–, CB–, 6.5–7.9 × 5.6–6.8 µm, L = 7.2 µm (σ = 0.44), W = 6.2 µm (σ = 0.39), Q = 1.17 (σ = 0.08) (*n* = 43/2).

**Habitat**—The species is commonly associated with angiosperm hosts, including southern beech (Nothofagaceae), but several collections were obtained from rotten trunks of *Pinus radiata*.

**Distribution**—Distributed throughout New Zealand but perhaps more common in the south.

**Additional specimen examined (paratype)**—New Zealand. South Island, West Coast Region, Lake Brunner, on an unidentified rotten branch, −42.620192°, 171.501389°,11 May 2018, J. Davies (PDD 112654).

***Pseudohydnum sinogelatinosum*** Y.C. Dai, F. Wu & H.M. Zhou, sp. nov. Figure 4.

Mycobank number: 844489.

**Type**—China. Yunnan Province, Lijiang, Ninglang County, Luguhu Nature Reserve, on stump of *Pinus*, 27.522500°, 100.738889°, elev. 3180 m, 9 September 2021, Dai 23017 (holotype, BJFC037590).

**Etymology**—***Sinogelatinosum*** (Lat.): refers to the species being similar to *Pseudohydnum gelatinosum* but occurring in China.

**Diagnosis**—Differs from other *Pseudohydnum* species by lateral stipitate basidiomata, pinkish buff to cinnamon-buff pileal surface when fresh, hymenophore with spines 3–4 per mm at base, strongly flexuous and branched hyphidia, growth on gymnosperm wood at high altitudes (>3000 m), and occurrence in Southwest China.

**Basidiomata**—Annual, gelatinous when fresh, brittle when dry, usually solitary, with a lateral stipe. Pilei applanate, occasionally lobed with an undulating margin, projecting up to 4.7 cm, 2.9 cm wide, 1.2 mm thick when dry. Pileal surface pinkish buff (5A3) to cinnamon-buff (4/5B4) when fresh, becoming milk-coffee to cigar-brown when dry. Spines white and conical when fresh, cream when dry, 3–4 per mm at base, up to 1 mm long when fresh. Context translucent when fresh. Stipe concolorous with pileal surface, shrinking to the base, translucent when fresh, up to 3.2 cm long and 4 mm in diam. when fresh.

**Hyphal structure**—Monomitic; generative hyphae with clamp connections. Contextual hyphae hyaline, thin- to slightly thick-walled, frequently branched, interwoven, 3–15 μm in diam. Tramal hyphae hyaline, thin-walled, frequently branched, interwoven, 2–3 μm in diam. Hyphidia strongly flexuous, occasionally branched. Basidia occasionally four-celled, barrel-shaped, globose to subglobose or ovoid, bearing guttules, 12–15 × 10–12 μm; sterigmata up to 8.5 μm long and 2.5–3 μm in diam. Probasidia fusiform to lageniform, 16.5–20 × 6.5–9 μm. Basidiospores broadly ellipsoid, ovoid to subglobose, hyaline, thin-walled, with a big guttule, IKI–, CB–, 7–9 (–9.2) × (5.2–)6–7.2 (–8) μm, L = 7.74 μm, W = 6.4 μm, Q = 1.16–1.24 (*n* = 90/3).

**Habitat**—The species is rather common on rotten gymnosperm wood at altitudes higher than 3000 m.

**Distribution**—Distributed in mountain areas with coniferous forests of Southwest China.

**Additional specimens examined (paratypes)**—China. Sichuan Province, Luding County, Hailuogou Forest Park, on dead tree of *Abies*, 29.574167°, 101.985556°, elev. 3000 m, 8 October 2021, Dai 23133 (BJFC037704); Jiulong County, Wuxuhai Forest Park, on fallen trunk of *Picea*, elev. 3200 m, 13 September 2019, Cui 17709 (BJFC034568); Yunnan Province, Lijiang, Yulong Snow Mountain, on fallen trunk of *Abies*, 27.253889°, 100.174722°, elev. 3200 m, 16 September 2018, Cui 17064 (BJFC030363); on fallen trunk of *Picea*, 16 September 2018, Cui 17074 (BJFC030373).

***Pseudohydnum tasmanicum*** Y.C. Dai & G.M. Gates, sp. nov. Figure 5.

Mycobank number: 844490.

**Type**—Australia. Tasmania, the Arve Loop Forest Reserve, on dead tree of *Eucalyptus*, −43.166667°, 146.808333°, 15 May 2018, Cui 16721 (holotype, BJFC030020).

**Etymology**—***Tasmanicum*** (Lat.): refers to the species being found in Tasmania, Australia.

**Diagnosis**—Differs from other *Pseudohydnum* species by pileate basidiomata with a rudimentary stipe base, a vinaceous gray to smoke-gray pileal surface when fresh, hymenophore with spines 2–3 per mm at base, broadly ovoid to subglobose basidiospores, growth on *Eucalyptus* and *Nothofagus*, and occurrence in Tasmania, Australia.

**Basidiomata**—Annual, pileate, with a rudimentary stipe base, a few imbricate, gelatinous when fresh, brittle when dry. Pilei dimidiate, shell-shaped, projecting up to 1.1 cm, 1.8 cm wide and 1.2 mm thick when dry. Pileal surface light vinaceous gray (13B2/3) to smoke-gray (F34) when fresh, become mouse-gray when dry. Spines white and conical when fresh, become buff when dry, 2–3 per mm at base, up to 1 mm long when dry. Context translucent, vinaceous buff when dry.

**Hyphal structure**—Monomitic; generative hyphae with clamp connections. Contextual hyphae hyaline, thin- to slightly thick-walled, occasionally branched, interwoven, 3–8 μm in diam. Tramal hyphae hyaline, thin-walled, usually branched, interwoven, 2–3 μm in diam. Hyphidia simple, hyaline. Cystidia absent. Basidia frequently four-celled, occasionally two-celled, barrel-shaped, ovoid to subglobose, bearing guttules, 12–15 × 10–11 μm; sterigmata up to 14 μm long and 2–3 μm in diam. Basidiospores broadly ovoid to subglobose, hyaline, thin-walled, occasionally with a guttule, IKI–, CB–, 7.2–9 (–9.2) × (5.7–)6–7.2 (–8) μm, L = 8.19 μm, W = 6.49 μm, Q = 1.23–1.3 (*n* = 60/2).

**Habitat**—The species was found on a fallen angiosperm trunk and dead standing angiosperm trees, e.g., *Eucalyptus* and *Nothofagus.*

**Distribution**—So far known from Tasmania, Australia.

**Additional specimen examined (paratype)**—Australia. Tasmania, Mount Field Forest, close to Mount National Park, on fallen trunk of *Nothofagus*, −42.683333°, 146.700000°, 14 May 2018, Dai 18724 (BJFC027192).

***Pseudohydnum totarae*** (Lloyd) J.A. Cooper, comb. nov. Figure 6.

Mycobank number: 844491.

**Basionym**—*Auricula totarae* Lloyd, Mycol. Writ. (Cincinnati) 6 (Letter 62): 935 (1920).

**Type**—New Zealand. North Island, Wellington Region, Weraroa, on rotting wood of *Podocarpus totara*, 12 July 1919, G.H. Cunningham (holotype BPI 701378, isotype PDD 225).

**Etymology**—***Totarae*** (Lat.): refers to the substrate, *Podocarpus totara*, of the type collection.

**Diagnosis**—Differs from other *Pseudohydnum* species by stipitate and confluent basidiomata, spathulate pilei, grayish brown to reddish brown upper surface when fresh, hymenophore with sparse spines (0.8–1.2 per mm at base), growth on gymnosperm wood, and known distribution restricted to New Zealand.

**Basidiomata**—Annual, gelatinous when fresh, stipitate, solitary to confluent at the base. Pilei spathulate 10–40 mm diam., pileal surface white to grayish brown (7E3–9E3) to reddish brown (11E5), smooth to minutely velutinate, with scattered, bluntly conical spines up to 0.2 mm long. Hymenophore white to pale-brown (8B2). Hymenophore spines white, conical, gelatinous, 1–3 mm long, around 1 mm diam. at base. Context translucent when fresh, 1–2 mm thick, concolorous with pileal surface. Pseudostipe lateral to eccentric, 20–50 mm long, and 8–14 mm in diam., cylindrical, sometimes laterally compressed, sometimes expanding towards the base and apex, concolorous with pileus, surface velutinate with blunt short spines similar to the pileal surface.

**Hyphal structure**—Monomitic; generative hyphae with clamp connections. Contextual and spine tramal hyphae hyaline, thin- to slightly thick-walled, frequently branched, agglutinate, 2–9 µm in diam. Basidia longitudinally cruciate-septate, 9–13 × 8 µm with four sterigmata up to 12 × 2 µm. Probasidia lageniform to clavate, with stalks 2–3 µm in diam. Hyphidia irregular, mostly unbranched, slightly swollen towards apex, clamped at the base. Basidiospores (excluding apiculus) broadly ellipsoid to subglobose, hyaline, thin-walled, occasionally with a guttule, germinating by repetition, IKI–, CB–, 5.5–6.5 × 4.8–5.7 µm, L = 6 µm, σ = 0.31; W = 5.2 µm, σ = 0.30; Q = 1.17, σ = 0.08 (*n* = 40/2).

**Habitat**—Originally described on rotten wood of Totara (*Podocarpus totara*). The species appears to be commonly associated with decaying gymnosperm wood; other recorded substrates include Kauri (*Agathis australis*) and Rimu (*Dacrydium cupressinum*).

**Distribution**—Throughout New Zealand but more common in the north and west in podocarp forests.

**Additional specimens examined**—New Zealand. North Island, Northland region, Trounson Kauri Park, on rotten wood of *Agathis autralis*, −35.72129000°, 173.64801000°, 11 May 2017, J.A. Cooper, JAC14517 (PDD 106891). Waikato Region, Pureora Forest, Rimu Track, on unidentified decorticate branch, −38.58341571°, 175.6337815°, 19 May 2011, J.A. Cooper, JAC11991 (PDD 96246). South Island, West Coast region, Moana, on unidentified rotten wood, −42.319907°, 171.478214°, 9 May 2018, N. Siegel, NS2672 (PDD 112652); Haupiri, on unidentified rotten wood, −42.572551°, 171.886597°, 11 May 2018, S. Carlson (PDD 112655).

**Notes**—The species was described as *Auricula totarae* by C.G. Lloyd (USA) in 1920 from material sent to him by C.G. Cunningham in New Zealand. The original description is sparse but refers to the protuberances on the upper surface of the fruiting bodies and the presence of microscopic globose bodies that Lloyd assumed to be basidia. The isotype consists of several small fruiting bodies and fragments. The material appears to be immature, but some fragments do bear the characteristic hymenophore spines of *Pseudohydnum* as well as the upper surface protuberances mentioned by Lloyd. The contemporary writings of Lloyd indicate that he was familiar with *Pseudohydnum gelatinosum* and we must assume that his placement of the species in *Auricula* was due to his having failed to observe the hymenophore spines. No mature basidia or spores were found in the isotype (PDD 225), although the immature globose basidia mentioned by Lloyd were abundant. Several of the fruiting bodies show a well-developed stipe. There is no doubt that this taxon is the same as the one re-described here based on modern taxonomy.

## 4. Discussion

There is unrecognized cryptic diversity within the genus *Pseudohydnum*, including three taxa detected from North America in our phylogenetic analysis (Figure 1). The different base pairs among these taxa account for >2% nucleotides in the ITS regions, and these taxa are treated as “*Pseudohydnum gelatinosum*-1”, “*Pseudohydnum gelatinosum*-2”, and “*Pseudohydnum gelatinosum*-3” in our study because we did not study the representative collections.

*Pseudohydnum gelatinosum* seems to be distributed in Eurasia (Table 1, Figure 1), and it has smaller basidiospores (5–6 × 4.5–5.5 μm) than those in *P*. *himalayanum* (7–8.5 × 6–7.2 μm), *P*. *orbiculare* (6.5–7.9 × 5.6–6.8 µm), *P*. *sinogelatinosum* (7–9 × 6–7.2 μm), and *P*. *tasmanicum* (7.2–9 × 6–7.2 μm) [3]. *Pseudohydnum gelatinosum* resembles *P*. *totarae*, sharing similar spore dimensions (5–6 × 4.5–5.5 μm vs. 5.5–6.5 × 4.8–5.7 µm), but the former has dense spines (5–7 per mm at base vs. 0.8–1.2 per mm at base).

*Pseudohydnum himalayanum*, *P*. *sinogelatinosum*, *P*. *orbiculare*, and *P*. *tasmanicum* have similar basidiospore dimensions. The former two species have dense spines (3–6 per mm at base) and are boreal species found at altitudes higher than 3000 m in Southwest China. The latter two species have sparse spines (0.5–3 per mm at base) and are temperate species found in Oceania.

*Pseudohydnum himalayanum* and *P*. *sinogelatinosum* have an overlapping distribution in Southwest China, but the former has 5–6 spines per mm at base and unbranched hyphidia, while the latter has 3–4 spines per mm at base and branched hyphidia.

*Pseudohydnum tasmanicum* and *P*. *orbiculare* are closely related phylogenetically (Figure 1) and share pileate basidiomata with circular pilei and no stipe, but *P. tasmanicum* has a distribution in Tasmania, an island in the far south of Australia, while the distribution of *P*. *orbiculare* is restricted to New Zealand. In addition, *P*. *tasmanicum* has a vinaceous gray to smoke-gray pileal surface, a hymenophore with 2–3 spines per mm at base, while *P*. *orbiculare* has a dark-gray, brown to black pileal surface when fresh and a hymenophore with sparse spines (0.5–1 per mm at base).

The two New Zealand species of *Pseudohydnum* have a similar size and identical range of coloration, from translucent white to gray or brown to reddish brown. They are distinguished by different basidiomata morphology; being spathulate with a pseudostipe in *P*. *totarae*, and orbicular to fan-shaped and without a pseudostipe in *P*. *orbiculare*. In addition, the former species prefers gymnosperm hosts, especially podocarp wood, and the latter is more common on angiosperm hosts, although the distinction is not absolute because of confirmed records of *P. orbiculare* on rotten wood of *Pinus radiata*, which is an exotic species in New Zealand. *Pseudohydnum orbiculare* is closely related to *P*. *tasmanicum* from Australia. ITS sequences show a consistent difference of 1.8%. This pair of species exhibit few or no morphological or ecological differences, which is probably a consequence of a relatively recent dispersal event with subsequent genetic isolation and divergence or maybe when Gondwana split Tasmania off from New Zealand. They are nevertheless good independent phylogenetic species.

*P**seudohydnum gelatinosum* var. *bisporum* differs from our new taxa by producing two basidiospores [11,12]. *P**seudohydnum gelatinosum* var. *paucidentatum* is different from our new taxa by producing only a few spines [13].

*Pseudohydnum brunneiceps* differs from our new taxa by its vinaceous brown to fuscous pileal surface when fresh, distinctly confluent basidiomata, and growth on *Cryptomeria* in a subtropical area of China [8].

*Pseudohydnum translucens* has pure white basidiomata when soaked and dry and subglobose basidiospores measuring 4–5 μm [7]. Lloyd thought that *P*. *translucens* and *P*. *gelatinosum* belonged to different genera since *P*. *gelatinosum* has a homogeneous texture both in spine and context, while *P*. *translucens* has a heterogeneous texture and the hymenial layer is different from the context, although both are white [7]. Anyhow, *Pseudohydnum translucens* is easily distinguished from our new taxa by its shorter basidiospores (4–5 μm vs. 5.5–9 μm long in our new taxa).

Phylogenetically, *Pseudohydnum sinogelatinosum* is related to *P*. *gelatinosum* and “*P*. *gelatinosum*-1*”* from North America, but *P*. *sinogelatinosum* is readily distinguished from *P*. *gelatinosum* by bigger basidiospores (7–9 × 6–7.2 μm vs. 5–6 × 4.5–5.5 μm), sparser spines (3–4 per mm at base vs. 5–7 per mm at base), and branched hyphidia (unbranched hyphidia in *P*. *gelatinosum*) [3]. We did not study the samples of “*P*. *gelatinosum*-1” and cannot comment on them.

## Figures and Tables

**Figure 1 jof-08-00658-f001:**
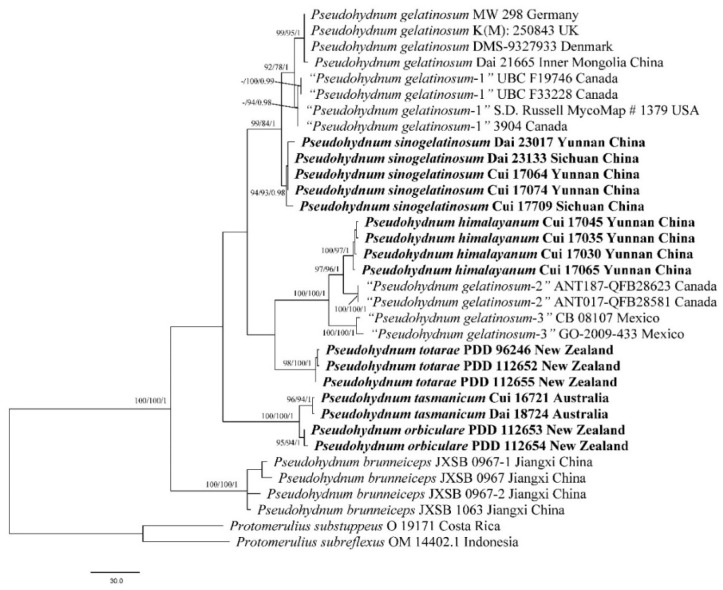
MP strict consensus tree illustrating the phylogeny of *Pseudohydnum* based on the concatenated ITS + nLSU dataset. Branches are labeled with bootstrap support for MP, ML, and BI results greater than or equal to 70% (MP-BS and ML-BS) and 0.90 (BPP). New taxa are in bold.

**Figure 2 jof-08-00658-f002:**
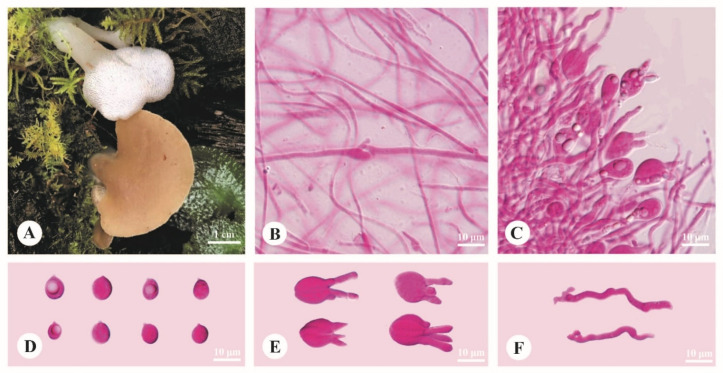
Basidiomata and microscopic structures of *Pseudohydnum himalayanum* (holotype, Cui 17045). (**A**) Basidiomata. (**B**) Tramal hyphae. (**C**) A section of hymenium. (**D**) Basidiospores. (**E**) Basidia. (**F**) Hyphidia.

**Figure 3 jof-08-00658-f003:**
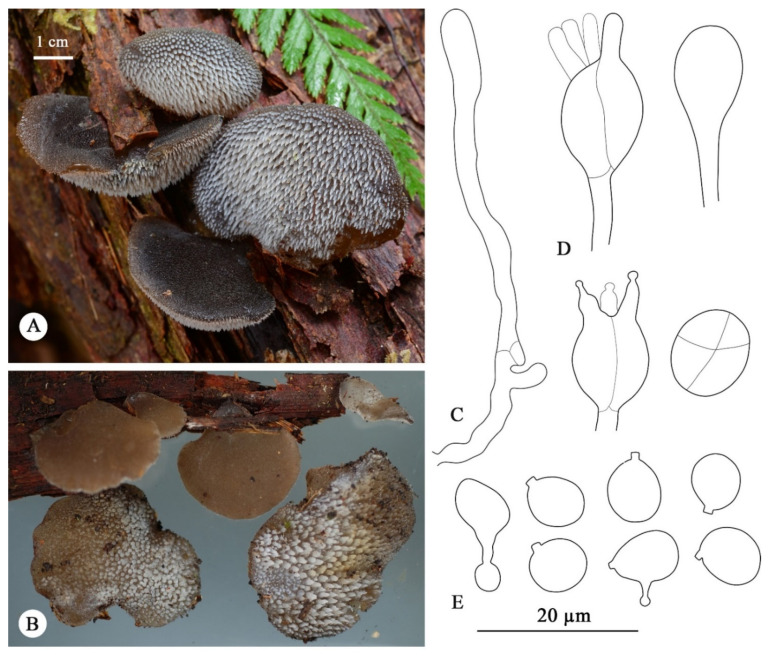
Basidiomata and microscopic structures of *Pseudohydnum orbiculare*. (**A**) Basidiomata (holotype PDD 112653). (**B**) Basidiomata (PDD 112654). (**C**) A hyphidium. (**D**) Basidia. (**E**) Basidiospores.

**Figure 4 jof-08-00658-f004:**
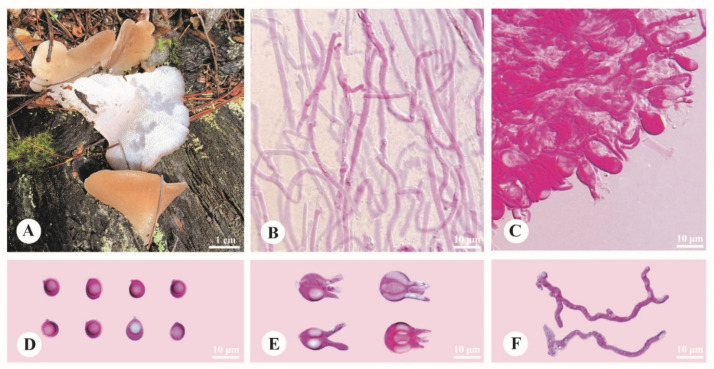
Basidiomata and microscopic structures of *Pseudohydnum sinogelatinosum* (holotype, Dai 23017). (**A**) Basidiomata. (**B**) Tramal hyphae. (**C**) A section of hymenium. (**D**) Basidiospores. (**E**) Basidia. (**F**) Hyphidia.

**Figure 5 jof-08-00658-f005:**
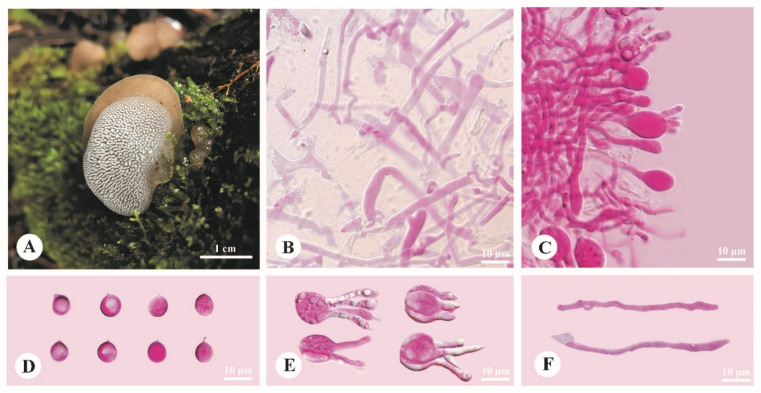
Basidiomata and microscopic structures of *Pseudohydnum tasmanicum* (holotype, Cui 16721). (**A**) Basidiomata. (**B**) Tramal hyphae. (**C**) A section of hymenium. (**D**) Basidiospores. (**E**) Basidia. (**F**) Hyphidia.

**Figure 6 jof-08-00658-f006:**
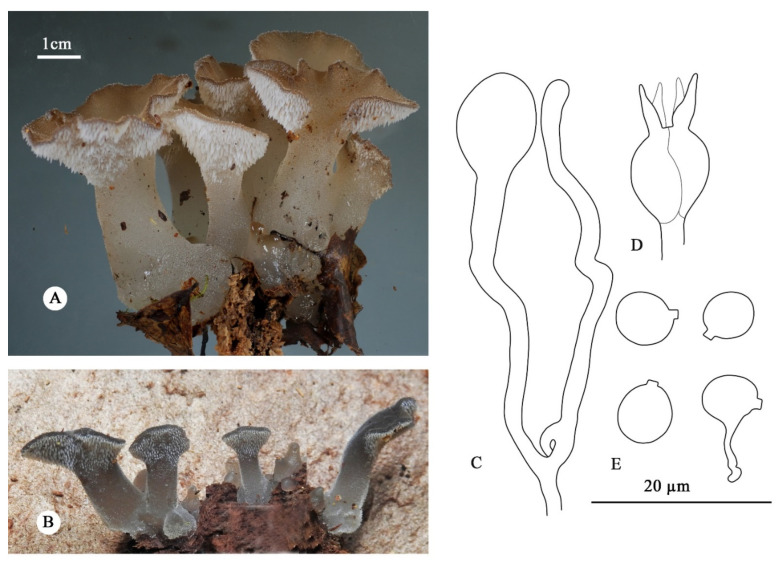
Basidiomata and microscopic structures of *Pseudohydnum totarae*. (**A**) Basidiomata (PDD 112655). (**B**) Basidiomata (PDD 106891). (**C**) A probasidium and a hyphidium. (**D**) A basidum. (**E**) Basidiospores.

**Table 1 jof-08-00658-t001:** Taxa information and GenBank accession numbers of the sequences used in this study.

Species	Location	Sample	ITS	nLSU
*Pseudohydnum brunneiceps*	Jiangxi, China	JXSB 0967	MN497254	MN497259
*Pseudohydnum brunneiceps*	Jiangxi, China	JXSB 0967-1	MN497255	MN497260
*Pseudohydnum brunneiceps*	Jiangxi, China	JXSB 0967-2	MN497256	MN497261
*Pseudohydnum brunneiceps*	Jiangxi, China	JXSB 1063	MN497257	MN497258
** *Pseudohydnum gelatinosum* **	**Inner Mongolia, China**	**Dai 21665**	**ON243826**	**ON243924**
*Pseudohydnum gelatinosum*	Denmark	DMS-9327933	MT644890	MT644890
*Pseudohydnum gelatinosum*	Germany	MW 298	DQ520094	DQ520094
*Pseudohydnum gelatinosum*	UK	K(M): 250843	MZ159722	-
“*Pseudohydnum gelatinosum*-1”	Canada	3904	KM406980	-
“*Pseudohydnum gelatinosum*-1”	Canada	UBC F19746	HQ604801	HQ604801
“*Pseudohydnum gelatinosum*-1”	Canada	UBC F33228	MG953967	-
“*Pseudohydnum gelatinosum*-1”	USA	S.D. Russell MycoMap # 1379	MK575262	-
“*Pseudohydnum gelatinosum*-2”	Canada	ANT017-QFB28581	MN992496	-
“*Pseudohydnum gelatinosum*-2”	Canada	ANT187-QFB28623	MN992495	-
“*Pseudohydnum gelatinosum*-3”	Mexico	CB 08107	KT875091	-
“*Pseudohydnum gelatinosum*-3”	Mexico	GO-2009-433	KC152166	-
** *Pseudohydnum himalayanum* **	**Yunnan, China**	**Cui 17030**	**ON243827**	**ON243925**
** *Pseudohydnum himalayanum* **	**Yunnan, China**	**Cui 17035**	**ON243828**	**ON243926**
** *Pseudohydnum himalayanum* **	**Yunnan, China**	**Cui 17045**	**ON243829**	**ON243927**
** *Pseudohydnum himalayanum* **	**Yunnan, China**	**Cui 17065**	**ON243830**	**ON243928**
** *Pseudohydnum orbiculare* **	**New Zealand**	**PDD 112653**	**ON243831**	**ON243929**
** *Pseudohydnum orbiculare* **	**New Zealand**	**PDD 112654**	**ON243832**	**-**
** *Pseudohydnum sinogelatinosum* **	**Yunnan, China**	**Cui 17064**	**ON243833**	**-**
** *Pseudohydnum sinogelatinosum* **	**Yunnan, China**	**Cui 17074**	**ON243834**	**ON243930**
** *Pseudohydnum sinogelatinosum* **	**Sichuan, China**	**Cui 17709**	**ON243835**	**ON243931**
** *Pseudohydnum sinogelatinosum* **	**Yunnan, China**	**Dai 23017**	**ON243836**	**ON243932**
** *Pseudohydnum sinogelatinosum* **	**Sichuan, China**	**Dai 23133**	**ON243837**	**ON243933**
** *Pseudohydnum tasmanicum* **	**Australia**	**Cui 16721**	**ON243838**	**ON243934**
** *Pseudohydnum tasmanicum* **	**Australia**	**Dai 18724**	**ON243839**	**ON243935**
** *Pseudohydnum totarae* **	**New Zealand**	**PDD 96246**	**ON243840**	**-**
** *Pseudohydnum totarae* **	**New Zealand**	**PDD 112652**	**ON243841**	**-**
** *Pseudohydnum totarae* **	**New Zealand**	**PDD 112655**	**ON243842**	**ON243936**
*Protomerulius subreflexus*	Indonesia	OM 14402.1	MG757508	MG757508
*Protomerulius substuppeus*	Costa Rica	O 19171	JX134482	JQ764649

New sequences are in bold.

## Data Availability

Publicly available datasets were analyzed in this study. The data can be found here: https://www.ncbi.nlm.nih.gov/; https://www.mycobank.org/page/Simple%20names%20search [51,52] (accessed on 22 April 2022).

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
