# Peer review of "Phylogeny and Diversity of the Genus *Pseudohydnum* (Auriculariales, Basidiomycota)"

_jof, 2022, doi:10.3390/jof8070658_

Round 1

Reviewer 1 Report

Some minor remarks and corrections:

Line 37 ‘var. bisporum Lowy & Courted’: correct Lowy & Courtec. (Courtecuisse)

Line 69 : ‘IKI =_Melzer’s_reagent’: Why not the abbreviation ‘MLZ’? 

IKI is not the same as Melzer’s reagent, however similar. See BARAL, H.O. (1987). Lugol's solution/IKI versus Melzer's reagent: hemiamyloidity, a universal feature of the ascus wall. ‑  Mycotaxon 29: 399‑450

Line 151 – 324: I miss a description of the observed colours (brown) on microscopic level: pigment incrusting or intracellular?

Line 307: ‘distribution restricted to New Zealand’- How can you be sure about this statement? Aren’t there more options (Papua New Guinea?)

Line 425: ‚Breitenbach, J.; Kränlin, F‘: correct … Kränzlin

Author Response

Thanks for your comments, we have carefully considered the suggestion by the reviewers and tried our best to improve this manuscript. Please see the follow:

Reviewer 2 Report

COMMENTS TO THE MANUSCRIPT: Phylogeny and diversity of the genus Pseudohydnum (Auriculariales, Basidiomycota) by Zhou et al.

General comment:

The submitted manuscript analyzes basidiocarp samples of the basidiomycete genus Pseudohydnum from China, New Zealand, and Australia. Based on detailed morphological (basidiomata and hyphidia shape, pileal surface color, spine and basidiospore size), host, biogeography and phylogenetic (ITS/LSU markers) analyses five new species are proposed.

The submitted manuscript is well written, and the information provided is the required for a new fungal species description. In general, all manuscript sections are clearly explained, and the new species descriptions are properly described and discussed. The subject of the submitted manuscript is of interest for mycologists, because adding species for the Fungal Kingdom aids to a better description on the diversity and complexity of such taxa. Thus, the manuscript is suitable to be published in the Journal of Fungi, after minor review. Below are some specific comments for the authors' consideration.

Specific comments:

1. Page 2, Line 54. The name of this section is wrong, it does not correspond to Molecular phylogenetics studies, but to sampling and morphological analysis. Please correct the subsection name.

2. It is desirable to provide the GPS data where the basidiocarp were collected. This kind of information is considered relevant when new species from new geographical locations are described, and geographic coordinates should be given as decimal degrees (Aime et al. 2021 IMA Fungus 12(1): 1-15). Please add such information in the corrected Sampling and morphological analysis subsection.

3. Page 4, lines 124-130. Please reallocate such data to the Materials and Methods section. I suggest numerating the 2.3 subsection as Molecular phylogenetics analysis, and properly relocate here these data.

4. Do the authors think it is possible to generate a Dichotomous key for Pseudohydnum genus, based on their detailed morphological analysis? Although not mandatory for manuscript publication, it will be useful for further studies on the genus.

5. Please do not forget to include TreeBase ID number and corresponding Mycobank numbers for all species in the final version of the manuscript.

Author Response

(The authors gave the same response as above.)

Reviewer 3 Report

Very well presented manuscript, with complete descriptions and excellent images. It could be suggested to unify the descriptions of the species and add the color codes in some descriptions, but they are only suggestions, minor changes that do not affect the quality of the work.

The inclusion of a key to the species worldwide is recommended, the information is already explained in the discussion, but its inclusion will facilitate the work for the identification of the species.

Author Response

Dear editor and reviewers

All the comments have been responded.  Please see the attachment. 
